# Nickel-organo compounds as potential enzyme precursors under simulated early Earth conditions
Philippe Diederich[1], Christian Seitz[2], Lance Buckett[1], Liesa Salzer[1], Thomas Geisberger[2], Wolfgang Eisenreich [2], Claudia Huber [2] & Philippe Schmitt-Kopplin [1,3,4] ✉

The transition from inorganic catalysis through minerals to organic catalysis by enzymes is a necessary step in the emergence of life. Our work is elucidating likely reactions at the earliest moments of Life, prior to the existence of enzymatic catalysis, by exploring essential intersections between nickel bioinorganic chemistry and pterin biochemistry. We used a prebiotically-inspired acetylene-containing volcanic hydrothermal experimental environment to shed light on the efficient formation of nickel-organo complexes. The simplest bis(dithiolene)nickel complex $(C_2H_2S_2)_2Ni$ was identified by UV/Vis spectroscopy, mass spectrometry, nuclear magnetic resonance. Its temporal progression and possible function in this simulated early Earth atmosphere were investigated by isolating the main bis(dithiolene)nickel species from the primordial experimental setup. Using this approach, we uncovered a significant diversity of nickel-organo compositions by identifying 156 elemental annotations. The formation of acetaldehyde through the subsequent degradation of these organo-metal complexes is intriguing, as it is reminiscent of the ability of *Pelobacter acetylenicus* to hydrate acetylene to acetaldehyde *via* its bis(dithiolene)-containing enzyme acetylene hydratase. As our findings mechanistically characterize the role of nickel sulfide in catalyzing the formation of acetaldehyde, this fundamental pre-metabolic reaction could play the role of a primitive enzyme precursor of the enzymatic acetylene metabolism and further strengthen the role of acetylene in the molecular origin of life.

Initially, bis(dithiolene)nickel compounds garnered attention solely for their synthetic properties. However, their resemblance to enzyme catalytic centers was later recognized, leading to renewed attention for those nickel-organo compounds. The synthesis of thiophene from acetylene and pyrite (FeS$_2$) by Steinkopf and Kirchhoff[1] originally inspired Schrauzer and Mayweg to investigate the reactions between diphenyl acetylene and nickel sulfides in toluene[2]. Schrauzer's discovery of bis(diphenyldithiolene)nickel, a nickel-sulfur complex with highly covalent metal-ligand bonds, led to the successful synthesis of various derivatives of this compound class, including the synthetically challenging unsubstituted complex $(C_2H_2S_2)_2Ni$, obtained from halogenated non-gaseous precursors and dissolved nickel ions[3].

Several extant enzymes possess metal-dithiolene active cores binding to transition metals like tungsten or molybdenum[4]. One example is acetylene hydratase (AH) expressed by the anaerobic bacterium *Pelobacter acetylenicus*. This organism is frequently discussed in the context of the origin of life because of its unique ability to hydrate acetylene into acetaldehyde, leading to acetyl-CoA and acetate building blocks. Hyperthermophile Archaea thriving in the vicinity of volcanic hydrothermal vents express aldehyde ferredoxin oxidoreductase (AFO) with a tungstopterin center catalyzing the oxidation of aldehydes into carboxylic acids at 100 degrees Celsius (C°)[4]. Interestingly, the catalytic ability of enzymes relying on metal-dithiolenes can be enhanced by exchanging molybdenum with

[1]Helmholtz Munich, Research Unit Analytical BioGeoChemistry, Neuherberg, Germany. [2]Technical University of Munich, TUM School of Natural Sciences, Department of Bioscience, Bavarian NMR Center (BNMRZ), Structural Membrane Biochemistry, Lichtenbergstr. 4, 85748 Garching, Germany. [3]Comprehensive Foodomics Platform, Chair of Analytical Food Chemistry, TUM School of Life Sciences, Technical University of Munich, Maximus-von-Imhof-Forum 2, 85354 Freising, Germany. [4]Max Planck Institute for Extraterrestrial Physics, Center for Astrochemical Studies, Gießebachstraße 1, 85748 Garching bei München, Germany. ✉e-mail: schmitt-kopplin@helmholtz-muenchen.de

nickel in molybdopterin-like catalysts[5]. Regardless of the presence of an enzyme structure, the nickel bis(dithiolene) complex alone is an excellent electron transporter as it exists in three oxidation states (0, −1, −2). As a result, these complexes activate hydrogen more efficiently than most hydrogenases in Nature[6], which aligns with the abiogenesis theory of pre-biotic synthesis of organic molecules.

The role of acetylene as a building block of terrestrial life remains a topic of speculation. Still, there is growing evidence for its presence in the atmosphere of interstellar bodies such as Titan[7] or Jupiter[8]. Additionally, its existence on early Earth is hypothesized[9,10], emphasizing the importance of investigating the role of acetylene in the emergence of life. Previous hypotheses regarding the role of acetylene in the origin of life have suggested that it contributes to the formation of polycyclic aromatic hydrocarbons[11] and small biomolecules[12]. Its popularity increased after it was proposed as a "fast food" for organisms in Earth's early biosphere[9]. Recent literature analyzed the potential of acetylene under the same conditions to form distinct unsaturated fatty acids in combination with carbon monoxide[13]. Moreover, a route to pyrrole was demonstrated in the presence of ammonia, leading to the elemental unit of porphyrin, another metal complexing scaffold indispensable for life as we know it today. Acetylene, combined with nickel sulfide, forms a large diversity of prebiotic organic compounds, including acetaldehyde and alanine, in a one-pot setup[14].

Nickel plays a fundamental role in enzymes found in most organisms, independent of the bis(dithiolene) framework. Nickel sites in extant enzymes show a large variety in nickel coordination and redox chemistry over a potential range of 1500 mV in proteins[15]. The metal center in superoxide dismutase (SOD) catalyzes redox processes with potentials ranging from +890−−160 mV[16]. In methyl-coenzyme M reductase (MCR) and carbon monoxide dehydrogenase (CODH), the active sites reach potentials as low as −600 mV[17].

Anaerobic nickel-containing CODH isoforms are found in organisms capable of heterotrophy, autotrophy, fermentation, or anaerobic respiration[18,19]. In hydrogenogenic CO oxidizers, CO oxidation is coupled with proton reduction to produce hydrogen $(H_2)$[20]. In the presence of water, required as an electron acceptor, a hydrogenase, and an ATP synthase are sufficient to ensure respiration. This straightforward mechanism highlights the significance of nickel-based catalysis in ancient energy preservation[21].

Nickel sulfide plays a crucial role in recent prebiotic chemistry, as they were shown to promote multiple relevant reactions. Thioesters were synthesized on nickel sulfide through geoelectrochemical $CO_2$ fixation[22]. Another study reported the promotion of a protometabolism in an alkaline hydrothermal vent environment[23]."

Ni-organo catalysts are pivotal in various organic reactions, including cross-coupling reactions (e.g., Suzuki, Kumada, or Negishi couplings)[24], carbon-heteroatom bond-forming[25] reactions and the hydrogenation of organic compounds[26]. Our experimental setup finds its roots in the "Iron-sulfur world" theory of Wächtershäuser who considered nickel or iron sulfide salts as catalytically active species due to their redox potential and limited solubility in water, which allows surface catalysis.

We now further investigated the previously mentioned one-pot setup[14] with a focus on the formation mechanism of acetaldehyde and its link to the presence of nickel-organo compounds. The one-pot setup already showed a progressing system that generated acetaldehyde over time and also changed its pH in the process due to formed acids, ultimately leading to a changed reaction path for the formed acetaldehyde. Building on the knowledge about the presence of acetaldehyde, we provide the first mechanistic evidence of abiogenic acetylene processing based on nickel bis(dithiolene), a molecular structure reminiscent of active centers of contemporary enzymes. Considering the resemblance with molybdopterin coenzymes, we postulate that nickel bis(dithiolene) was a transition catalyzer, allowing early acetaldehyde formation from acetylene in a prebiotic environment.

## Results

### Formation and identification of nickel-organo complexes

The investigated prebiotic system consisted of an oxygen-free carbon monoxide and acetylene atmosphere in equal amounts in the headspace of water containing water-insoluble nickel sulfide salt (NiS). The setup was contained in a sealed glass bottle and was incubated at 105 °C for varying periods of time.

The presence of nickel complexes was detected using Fourier-transform ion-cyclotron-resonance mass spectrometric (FT-ICR-MS) measurements after 2 h of incubation. FT-ICR-MS analysis allowed a reliable elemental annotation of the nickel-organo complexes in the complex reaction mixture. $C_4H_4S_4Ni$ was the most abundantly detected nickel-

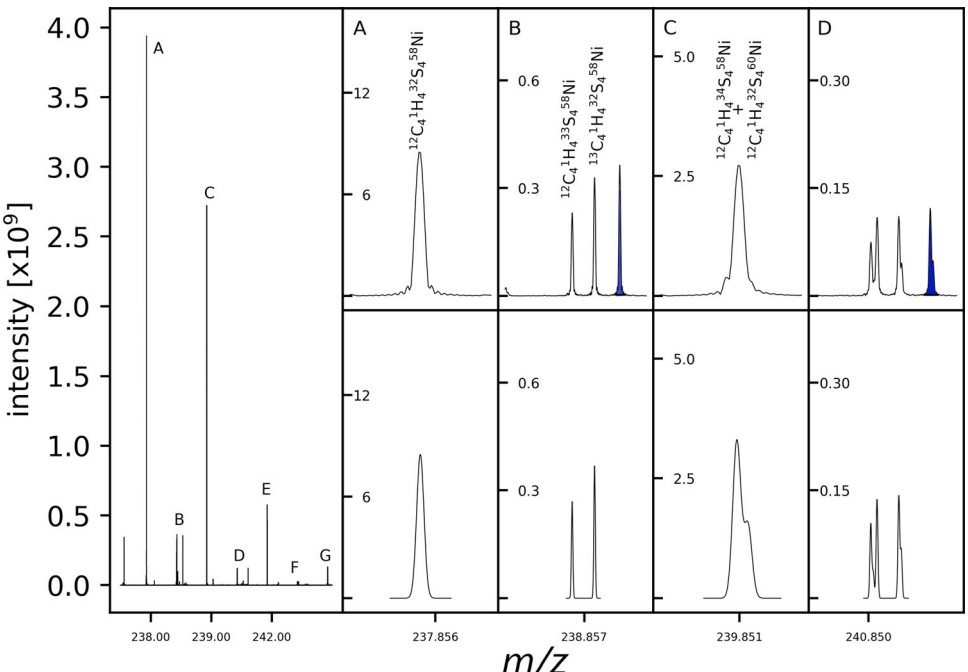

**Fig. 1 | True isotopic pattern of nickel bis (dithiolene).** FT-ICR-MS mass spectrum (left) with its annotated isotope pattern signals (top right) in comparison to the theoretical pattern (bottom right) enlarged in (**A**–**C**). **D** shows a further cluster of isotope signals made up of permutations of the isotopes in (**A**–**C**). Additional signals in blue highlight the -2 charge state. Signals marked **E**–**G** are further isotopic patterns belonging to the compound not shown in a zoomed in panel.

organo species by intensity. This annotation fits the simplest dithiolene-nickel species that can be obtained from two acetylene molecules, four sulfur, and one nickel atom. Due to its high abundance in the system, the composition could be further validated with its isotopic fine structure, excluding the possibility of false elemental annotation due to an isobaric compound (Fig. 1).

Samples showing the highest intensity of this compound in MS-spectra display a pink color (Suppl. Fig. 1). This color and the mass spectra signals for $C_4H_4S_4Ni$ decreased over time and were lost after 3 days of incubation.

A closer inspection of the isotope pattern revealed a further signal (Fig. 1, B+D highlighted in blue) not explained by the theoretical simulated isotope pattern. The mass difference between this additional peak and the isotope peak of the previous nominal mass amounts to the mass of one hydrogen atom. This result can be explained by the different possible oxidation states of nickel-bis(dithiolenes). The main peak belongs to the $M^{-1}$ species. The additional peaks belong to the $[X^{-2} + H^{+1}]^{-1}$ species, with X being any isotopic species with a sufficiently high intensity to detect the second oxidation state species. The same observation was made for sodium ions but with lower signal intensity.

UV/VIS spectroscopy of the colored reaction product in water showed two absorption maxima at 500 and 855 nm. The second maximum is in the near-infrared at 855 nm. Absorption in the near-infrared is characteristic of bis(dithiolene)nickel compounds[27]. The maximum at 500 nm explained the observed pink color as absorption of green light results in red transmitted light and is also expected to be associated with the detected complex. Degradation of the complex in water happened relatively fast, and the corresponding UV/Vis signals nearly disappeared completely after 4 h at room temperature (Fig. 2, A). The degradation of the compound was accompanied by a color change of the solution from pink to yellow.

### Isolation of the nickel-organo complex

We isolated the described complex from the aqueous solution *via* liquid-liquid extraction using methyl tert-butyl ether (MTBE). The MTBE phase took on a strongly violet color after the extraction of the aqueous phase. The MTBE phase was completely dried under $N_2$ flow, and a dark blue solid remained in the glass vial. This color is also reported by Schrauzer, who synthesized the compound through a different non-aqueous route with halogenated reactants, incompatible with currently hypothesized conditions on early Earth[3]. We weighed the remaining solid of two different experimental setups in triplicates, differing in the composition of the gaseous atmosphere (Table 1). The setup with carbon monoxide produced 5.97 mg of the complex with a theoretical yield of 1.79%. In contrast, a setup with twice the volume of acetylene in the absence of carbon monoxide generated 6.92 mg with a yield of 1.14%. Utilizing twice the volume of acetylene increased the absolute yield as expected but decreased the theoretical yield. These yields are comparable to those reported by Schrauzer, who obtained a 3% yield using a more elaborate synthetic route. One triplicate of the CO/acetylene setup only showed very faint coloration. This triplicate also

showed the lowest residual mass. A reaction scheme and discussed compounds can be found in the supplementary information (Suppl. Fig. 2).

The near insolubility of the isolated compound at room temperature in purified water strengthened our hypothesis of a changed charge state. Even though the color change could result purely from the solvent change, another explanation is the change of the complexes' oxidation state from a charged water-soluble state to an uncharged lipophilic state. The solid only slowly dissolved at temperatures >50 °C, resulting in the expected pink coloration, followed by rapid degradation. Furthermore, adding the original reaction medium did not solubilize the compound at room temperature, excluding the possibility of a solubility-enhancing effect through molecules or salts formed in the experimental setup. Extraction of the residual solid phase in the reaction bottle with MTBE, expected to be mainly solid nickel sulfide salt, did not yield any amount of the target compound (no coloration of the MTBE used for extraction). We concluded that the compound remained solubilized during the experiment without precipitation and changed its oxidation state during extraction.

The isolated compound was further analyzed *via* LC-MS/MS to validate the hypothesized structure. LC-MS analysis of the dried extracted solid dissolved in methanol resulted in a violet solution with the same appearance as the MTBE solution. We detected the expected mass previously found in the initial mixture identified as $(C_2H_2S_2)_2Ni^-$. Fragmentation was performed with a UHPLC-IM-Q-ToFMS system to gather fragmentation data (Fig. 2, B) and report its collision cross section (CCS) value of 130 Å$^2$.

### Fate of the formed nickel complex

The isolated water-insoluble complex became soluble at around 50 °C and above. Dissolution of the complex also resulted in a color change back to pink, comparable to the color of the unextracted aqueous mixture, reflecting the presence of the charged state in water.

Incubation of the isolated complex in water at pH 11 for 15 and 30 h resulted in the degradation of the complex. The degradation route and degradation products are temperature-dependent. The complex degraded into formic acid and acetaldehyde at temperatures reflecting the initial experimental conditions (100 °C) (Fig. 3). This result confirms the previously observed characteristics of those two compounds in the system and sheds light on the exact mechanism of their formation with nickel sulfide[11]. Incubation at 24 °C yielded no acetaldehyde. NMR measurements of the incubated sample showed increased signals in the olefinic ppm range. This observation shows the formation of new carbon-carbon bonds via this complex during its degradation. The complex degraded into unsaturated acetylene polymers. NMR spectra can be found in Supplementary data 1.

In the degradation process, nickel sulfide is consistently formed. The identification was made by observing an insoluble black solid in the reaction vessel. However, likely, the nickel and sulfur released during degradation precipitated at least partially into nickel sulfide. This particular process provides the complex with the ability to dissolve nickel sulfide, making it

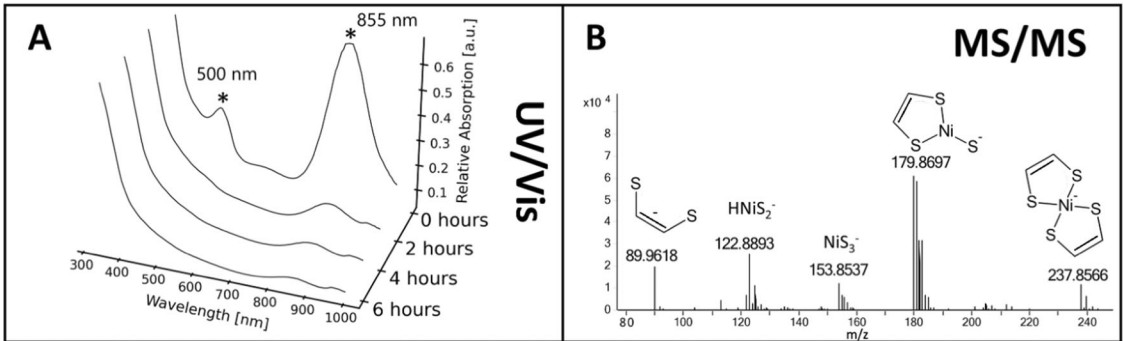

**Fig. 2 | Identification of bis(dithiolene) nickel. A** UV/Vis spectrum of the complete aqueous 2 h reaction product degrading over time. **B** MS/MS of the m/z 237.85 with suggested structures for the detected fragments.

**Table 1 | Nickel bis(dithiolene) yield.**

| Gas mixture | Gas Ratio (%) | All yields (mg) | Mean Yield (mg) | Theoretical yield (%) | Relative Standard deviation (%) |
|---|---|---|---|---|---|
| CO/acetylene | 50/50 | 5.97/5.98/(1.92)* | 5.97 | 1.79 | 0.1% |
| Acetylene | 100 | 6.81/7.19/6.76 | 6.92 | 1.14 | 3.4% |

Yields from triplicate experiments with two different atmospheres determined by weighing after drying. Gas ratio describes the ratio of the two gases used in the setups. Every yield is shown for each triplicate. The *marked triplicate was excluded from the yield determination.

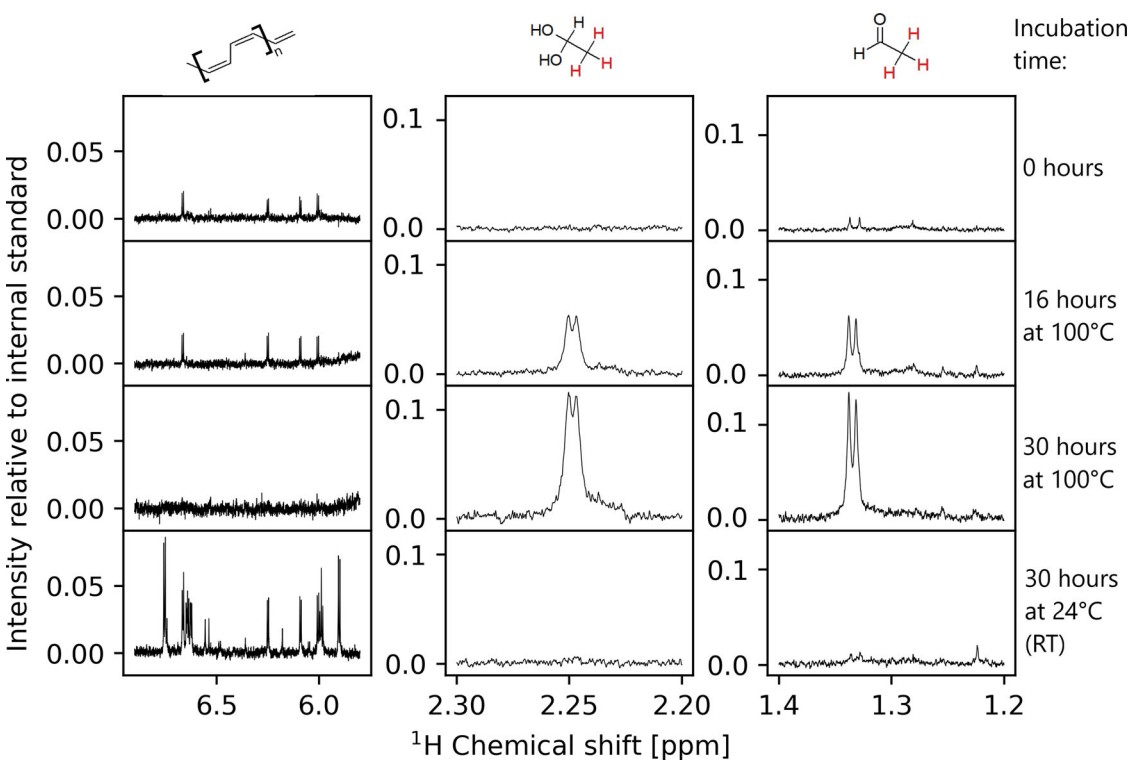

**Fig. 3 | Temporal progression of the isolated nickel complex over time.** The ppm range with the main changes is zoomed in. Structures on top show the observed hydrogen in red for acetaldehyde. As the structure of the olefinic species could not be determined, only the structure of an olefinic chain is shown. The incubation time indicates the time the sample was kept at 100 °C or room temperature, respectively.

soluble, transport this highly insoluble salt and relocate it via diffusion processes.

### Identification and characterization of diverse nickel-organo species

The compound with the elemental composition $(C_2H_2S_2)_2Ni$ is not the only nickel-containing annotation in the mixture. 156 different elemental compositions were annotated, some containing various amounts of oxygen. The addition of oxygen to the sulfur atoms in those complexes was already described and discussed by Schrauzer[28]. The detected diversity of nickel-organo compounds is vast and sometimes challenging to interpret, as some elemental compositions contain odd amounts of carbon atoms, a fact hinting at a carbon cleavage pathway with nickel-organo compounds as intermediates. This hypothesis is also strengthened by the detection of formic acid after the degradation of the complex. Most annotations show the $NiS_4$-core of the bis(dithiolene)nickel complex, followed in number by $NiS_3$-"cores" (Fig. 4). It cannot be excluded that some or all $NiS_3$-annotations result from in-source fragmentation, as the fragmentation pattern of $[(C_2H_2S_2)_2Ni]^-$ also showed a fragment with the formula $[C_2H_2S_3Ni]^-$. Nickel-organo annotations become increasingly scarce with higher amounts of sulfur atoms in the molecule. The $NiS_6$-"core"-group is nearly devoid of signals. Only the $[C_6H_xNiS_6]^-$ compositions (highlighted in yellow in Fig. 4) show some diversity in their homologous hydrogen series and could belong to the trimer, carrying an additional dithiolene unit.

### Discussion

The formation of the reported bis(dithiolene) nickel complex after 2 h of incubation of the evolving reaction mixture showed the feasibility of its existence in early earth scenarios. The similarity of the formed bis(dithiolene) nickel complex with the active center of AH found in *Pelobacter acetylenicus* living in anaerobic aqueous systems is very intriguing, as the setting of the experiment is situated in a prebiotic oxygen-free atmosphere. Acetaldehyde is obtained from acetylene in the presence of NiS via the formed nickel bis(dithiolene) and then oxidized to acetic acid[14](Fig. 5). This oxidation corresponds to a reaction performed by the hyperthermophilic archaeon *Pyroccocus furiosus via* AFO, another molybdopterin-containing enzyme. The likeness of the habitat of this microorganism is even more striking, as it lives near hydrothermal vents at 100 °C.

The described transportation role of the complex is also worth mentioning. The ability to extract nickel sulfide from minerals, to be later redeposited after degradation, leads to new scenarios, where nickel-catalyzed reactions like the reduction of double bonds[29] can occur on primitive Earth. The change in polarity, from an ionic form into a highly lipophilic organic form, could also lead to enhanced complex uptake into vesicles with less permeable and hydrophobic properties. This aspect makes the reported complex not only relevant for early prebiotic stages but also for later stages during a transition from reactions in an aqueous bulk to an evolving metabolism in protocells. Its chemical nature facilitates its introduction into vesicular protocells. This hypothesis is strengthened by a

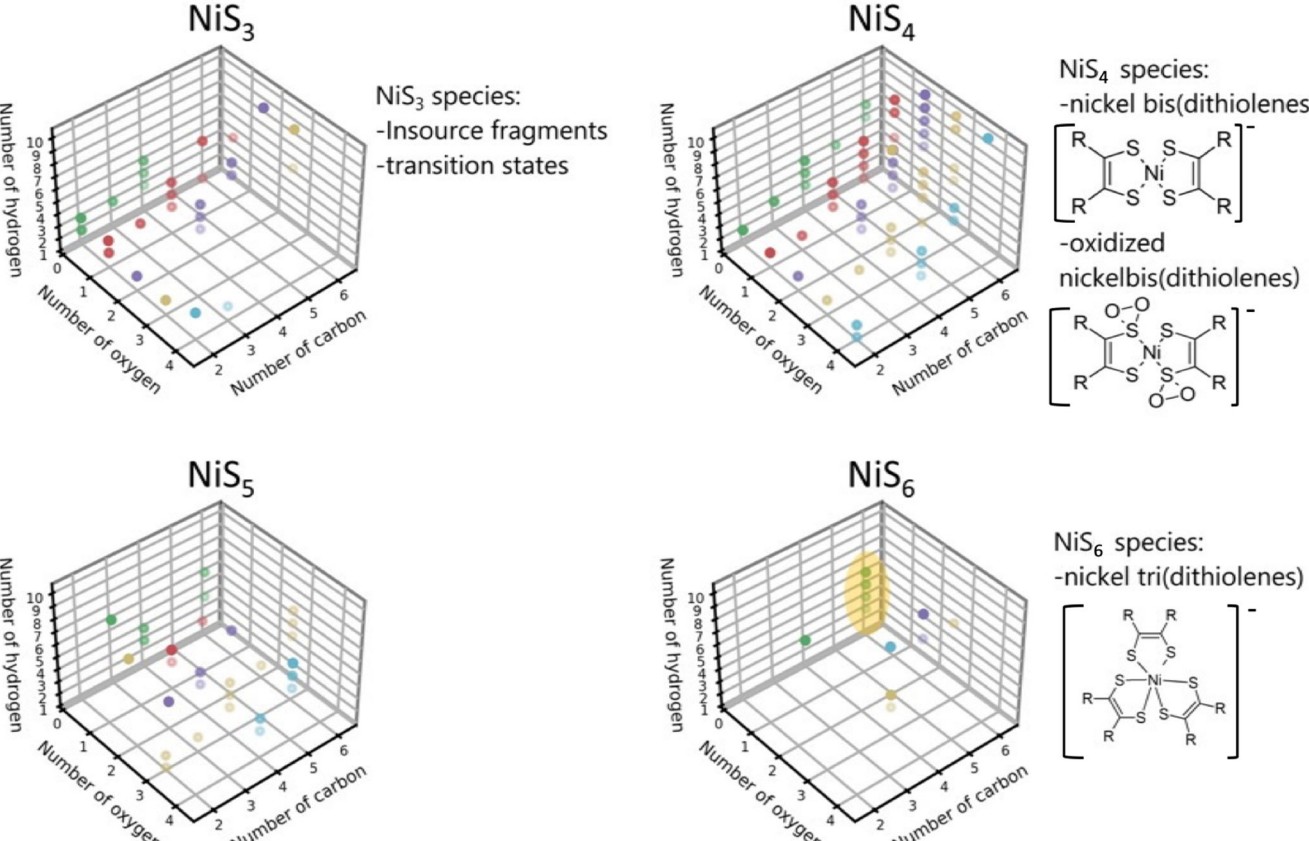

**Fig. 4 | Nickel-organo complexity.** 3 dimensional plots show the diversity in elemental composition for detected signals containing nickel and 3,4,5 or 6 sulfur atoms. Potential explanations for the $NiS_x$ motives are given next to the plots. $NiS_5$ annotations most likely belong to insource fragmentation, but their exact identity is not known. The color of the dot represents the number of oxygen (0 for green, 1 for red, 2 for violet, 3 for yellow and 5 for blue). A yellow ellipse highlights the only homologous hydrogen series for compounds with one nickel and six sulfur. Residuals marked as R consist of carbon, hydrogen, and/or oxygen but cannot be further characterized.

recently published report of vesicle formation in the described set-up, where the formed long-chain fatty acids showed the potential to form vesicles[30].

The described complexity of the nickel-organo complexes has the potential for the derivatization of this core motif for co-enzymes. Derivatization of the elemental nickel bis(dithiolene) alters its reactivity and potential to catalyze different reactions. The literature on nickel bis(-dithiolene) complexes provides a large list of catalyzed reactions involved in the energy management of microorganisms. The reduction of hydrogen atoms to form hydrogen gas is among the most interesting. Bis(dithiolene) nickel complexes lower the reduction potential required for proton reduction resulting from formed acids[6]. Ameerunisha et al. reported the reduction of protons from p-toluene sulfonic acid through a nickel bis(dithiolene) complex[6]. The fact that this environment produced a large diversity of unsaturated carboxylic acids from carbon monoxide and acetylene[13] increases the relevance of this mechanism, as those acids dissociate readily under aqueous conditions. The second origin of hydrogen cations in the system is acetic acid. Another reduction performed by those complexes is the reduction of $CO_2$ into formic acid[31]. The ability of nickel bis(dithiolene) to act as an electron transporter and to exist in several oxidation states lets it perform energy transfer roles.

The formation of carboxylic acids synergizes well with the redox reactions catalyzed by this compound class. Metal-bis(dithiolene) compounds are potent hydrogenases. Nickel bis(dithiolene) complexes can reduce protons from organic acids to hydrogen gas[6]. One characteristic of the investigated system is the production of many highly unsaturated acids *via* Reppe chemistry[13] and other acids. A compound reinforcing the suspected hydrogen activation is propanoic acid. Acrylic acid is the primary product of acetylene and carbon monoxide via the chemistry described by

Reppe in 1953[32]. Detection of its reduced form of propanoic acid[33] in the mixture indirectly indicates a possible reduction with hydrogen gas.

In addition to its redox capabilities, bis(dithiolene) nickel compounds act as intermediates for many organo-sulfur compounds[34]. One main follow-up product of bis(dithiolene) nickel is thiophene, proven to exist in large amounts in the investigated mixture[35].

Electron transfer is a vital mechanism of all living systems and is performed by NAD(H)/NADP(H) to allow ATP synthesis in extant organisms. The fact that acetylene is transformed to a large degree into aldehydes and additionally leads to a complex very similar to several active centers in enzymes connected to the metabolism of those compounds is a meaningful coincidence. A recent paper even showed its potential to oxidize glucose into gluconolactone[36]. Extant enzymes with pterin co-factors contain mostly molybdenum or tungsten. Yet nickel is a much more frequent element on Earth[37] with an elemental abondance of 1.86% compared to Mo with 2.35 ppm and W with 180 ppb; therefore, the early existence of precursors of active centers with nickel is a plausible hypothesis. Additionally, nickel is found in the active centers of 9 contemporary enzymes (known so far)[38]. Further studies have to be performed to investigate what prebiotic conditions allow for an exchange of nickel through molybdenum or tungsten."

Our findings revealed a pathway from acetylene to acetaldehyde via a nickel-organo complex. This discovery strengthens the relevance of nickel-organo compounds for the origin of life, as they now present an intermediate product for a prebiotic synthesis pathway of higher aldehydes, the amino acid alanine and potentially higher amino acids. Furthermore, this finding now paves the way for investigations about the exact catalytical nature and other roles in prebiotic chemistry of this nickel-organo compound.

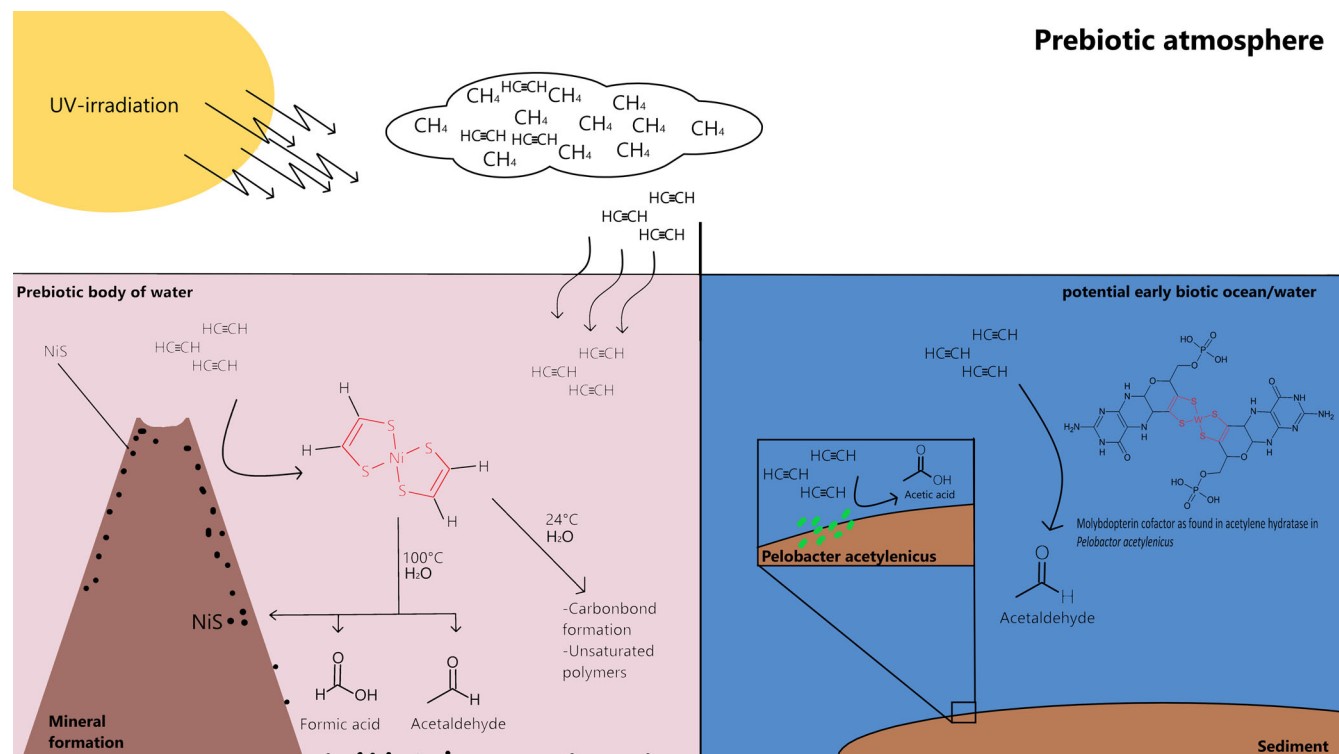

**Fig. 5 | Overview of a potential scenario forming acetaldehyde from acetylene via nickel bis(dithiolene). A** Reaction scheme showing a plausible pathway under early earth conditions leading to bis(dithiolene) nickel and its follow-up products. **B** Structure of the molybdopterin cofactor in Pelobacter acetylenicus and the enzymatic transformation of acetylene into acetaldehyde in a potentially more modern body of water that, however, still contains acetylene.

## Conclusion

This work showed the feasibility of forming bis(dithiolene) nickel under simulated early Earth atmosphere imitating hydrothermal volcanic conditions in the presence of acetylene. Even though the main nickel-organo compound $(C_2H_2S_2)_2Ni$ is nearly water-insoluble, its spontaneous self-assembly from acetylene allows its charged existence in water. We characterized the structure of the detected nickel bis(dithiolene). We showed an intriguing similarity between the transformation of acetylene to acetaldehyde by the tungstopterin co-enzymes and the formation of acetaldehyde *via* the degradation of the self-assembled nickel bis(dithiolene). The structural resemblance between bis(dithiolene) and extant co-enzymes supports the hypothesis that metal bis(dithiolenes) may have been among the first protoenzymes. Altogether, our work explores an essential intersection between nickel bioinorganic chemistry and pterin biochemistry, elucidating likely reactions at the earliest moments prior to the existence of enzymatic catalysis.

## Methods
### Bottle setup
A 125 ml glass serum bottle was charged with 1.0 mmol NiSO$_4$ • 6 H$_2$O (99%, Aldrich) and sealed with a silicon stopper. The bottle underwent three cycles of evacuation and argon filling, ultimately reaching a deaerated state. Subsequently, the bottle was filled with argon-saturated water (calculated for the end volume of 5 ml), with 1.0 mL argon-saturated 1 M Na$_2$S (solid Na$_2$S: 99.99%, Aldrich) solution, with 1.0 mL 1 M NaOH solution and finally with 60 ml CO and 60 ml acetylene (acetone-free), using gastight syringes for the injections. Reactions were carried out at 105 °C. Following a reaction time of up to 3 days, the reaction mixture was cooled down.

### FT-ICR-MS analysis
Samples were taken from the serum bottle with a syringe and centrifuged for 5 min at 15000 rpm. 100 µl of supernatant were diluted in 900 µl methanol and centrifuged to remove the precipitated salt. 70 µl of the centrifuged sample were diluted in 930 µl methanol to obtain the final sample dilution. Samples were measured in triplicates. All measurements were done in negative ionization mode.

Analysis was performed on a high-field Fourier Transform Ion Cyclotron Resonance mass spectrometer from Bruker Daltonics—Solarix. The magnet is a 12 T magnet from Magnex. The mass spectra were acquired with a 4 mega word (MW) time domain. The system was calibrated with L-arginine clusters in negative mode (5 mg L$^{-1}$ L-arginine solved in methanol). For each sample, scans were accumulated in the mass range of 122–1000 amu. Ions were accumulated for 300 ms. The pressure in the hexapole was $3 \times 10^{-6}$ mbar, and the pressure in the ICR vacuum chamber was $6 \times 10^{-6}$ mbar. An Apollo ii (Bruker Daltonics) ESI source was used. The supernatant was injected via a microliter pump system (flow rate: 120 µl h$^{-1}$).

Formula assignment was achieved through a network approach[39].

### LC-IM-Q-ToFMS analysis
LC-IM-MS/MS was carried out using a 6560 Ion Mobility LC/Q-TOF system (Agilent Technologies, Santa Clara, CA) coupled with a 1290 infinity II UHPLC (Agilent Technologies, Santa Clara, CA). Before the analysis, the MS was tuned within the parameters recommended by Agilent. The sample was injected onto a Waters ACQUITY UPLC BEH C8 (150 mm × 11 mm, 1.7 µm) column, with a starting gradient of 99% ACN (LC-MS/MS grade, Supelco LiChroSolv hypergrade for LC-MS) lowered linearly to 50% over 1 min with Milli-Q grade water and a total run time of 9 min at a flow rate of 0.2 mL/min. During the entire analysis, a reference mass mixture containing purine and hexakis(1H,1H,3H-tetrafluoropentoxy)phosphazene (Agilent Technologies, Santa Clara, CA) was introduced at a flow rate of 0.5 µL/min. The MS method used negative polarity with all ion fixed 2 ramped mode selected following drift time 0–59 with 10–40 CE in the second phase and a mass range from 50–1700. The gas temp was set at 250 °C with a flow of 12 L/min. The nebulizer was set at 40 psi, the sheath gas at 320 °C with a flow of 11 L/min, and vcap set at 5500 V. The IM measurements were conducted

in $N_2$ gas at ~4 Torr and 25 °C with a trap fill time of 2000 μs and release of 300 μs.

The raw file was calibrated to the reference mixture using an IM-MS reprocessor (Agilent Technologies, Santa Clara, CA), and the CCS was calibrated using a calibration mixture (Agilent Technologies, Santa Clara, CA) introduced to the IM-MS prior with a single field setting in Masshunter IM-MS browser (Agilent Technologies, Santa Clara, CA) result was applied to the sample measurement. Feature finding was carried out using an automatic and unbiased option in the mass profiler, and the CCS value resulted accordingly.

### NMR analysis

$D_2O$ (for analysis, Merck KGaA) was prepared with added sodium trimethylsilylpropanoate (2 mM) (98 atom %D, Aldrich) as the internal standard. This solution was used to reference and quantify signals in 1D-1H experiments. The pH of the sodium phosphate buffer was adjusted to 11 with sodium hydroxide. This solution was used to keep the dissolved complex at pH 11. The buffer was deoxygenated by nitrogen purging.

### Detection of degradation products

1D $^1H$ spectra were acquired in triplicates for the dissolved complex in deoxygenated buffered water (100 mM sodium phosphate at pH11) after 0, 15, and 30 h of incubation at 100 °C. The sample at room temperature was prepared identically and measured after 30 h. 150 μl dissolved complex were spiked with 40 μl phosphate buffer and 10 μl D2O containing 2 mM TSP. The pulse program consisted of a simple 90° pulse followed by acquisition. 8 scans were acquired for each sample with a relaxation delay of 26 s and an acquisition time of 4 s. On-resonance pre-saturation was used during the relaxation delay to suppress the water signal. The optimized 90° pulse had a duration of 10.35 μs. Relative quantification was done by comparing the sum of the integrals after baseline correction of the hydrate single hydrogen signal and the methyl hydrogens signal to the TSP internal reference integral. The average for every time point was calculated. The acquired FID was apodized with an exponential function (LB = 0.3), and Fourier transformed.

### Isolation of nickel bis(dithiolene)

We isolated the described complex from the aqueous solution *via* liquid-liquid extraction with methyl tert-butyl ether (MTBE). 9 mL of the reaction mixture was extracted with 3 × 9 mL of MTBE in a 50 ml falcon tube. The MTBE phase became strongly violet after the vortexing of the two-phase system. The MTBE phase was separated from the aqueous phase by carefully pipetting the MTBE off. The 3 × 9 ml of MTBE were combined and dried with an $N_2$ flow. After reducing the volume to about 1 ml, the residual sample was transferred to a 1.5 ml glass vial that was weighed beforehand. The residual sample was then dried to completeness and weighed to determine the amount of solid in the glass vial. A dark blue solid remained in the glass vial. This procedure was repeated three times for the two examined gas ratios.

### Data availability

The authors declare that [the/all other] data supporting the findings of this study are available within the paper [and its supplementary information files]. Further data that support the findings of this study are available from the corresponding author upon reasonable request. All NMR spectra can be found in Supplementary Data 1. Data for Fig. 4 can be found in Supplementary Data 2.

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

## Acknowledgements
This research was funded by the Deutsche Forschungsgemeinschaft (DFG, German Research Foundation) – Project-ID 364653263 – TRR 235 (CRC 235). CS and CH thank the Hans-Fischer-Gesellschaft (D-Munich) for financial support.

## Author contributions
PD conceived and designed the analysis, collected the data, performed experiments, performed data analysis, and wrote the manuscript. CS, LB, LS, TG, and CH contributed analysis tools. TG and CS performed experiments. LB, LS, WE, CH, and PSK edited the manuscript. PSK, CH, and WE provided funding and supervised the project.

## Funding

## Competing interests
The authors declare no competing interests.
