## [Peer Review File · Communications Chemistry]

Reviewers' comments:

Reviewer #1 (Remarks to the Author):

The paper by Diederich et al. reported the in-situ generation of Ni organo-compounds under simulated early atmospheric-ocean conditions. The generated products have a similar structure to enzymes, suggesting the geological origin of the Ni-containing enzyme. All the experiments, especially the characterization of the mixed products, were carefully performed and the interpretation of the data were solid. Considering the versatile catalytic and redox functions of Ni organo-compounds, the proposed model of abiotic formation of Ni catalysts is intriguing and will provide molecular basis understanding of the transition from geochemistry to more sophisticated chemical reactions catalyzed by proto-enzyme-like catalysts. This work also provides the new scientific bridge between pure inorganic chemistry in the laboratory and the prebiotic chemistry that drives the emergence of life on Earth. Given the novelty of this study, I support its publication in Communications Chemistry.

A weakness of this paper is the geological condition on which the authors stand. The authors refer to the iron-sulfur world hypothesis, which is the idea that metal-sulfide minerals at deep-sea hydrothermal vents played a crucial role in prebiotic chemistry. Meanwhile, the author's research also stands on atmospheric photochemistry to generate CO and acetylene. Considering the diffusion processes of small molecules, I found it difficult to consider that CO and acetylene are transported from the atmosphere to the deep-sea hydrothermal vents against the dilution processes in air and water. In this regard, the authors may consider the recent works on Ni-sulfide catalyzed CO₂ to CO conversion and subsequent formation of prebiotically important compounds at deep-sea hydrothermal vents.

N. Kitadai, R. Nakamura, M. Yamamoto, S. Okada, W. Takahagi, Y. Nakano, Y. Takahashi, K. Takai, Y. Oono, Thioester synthesis through geoelectrochemical CO₂ fixation on Ni sulfides, *Commun. Chem.*, 2021, 4, 37

N. Kitadai, R. Nakamura, M. Yamamoto, K. Takai, N. Yoshida, Y. Oono, Metals likely promoted protometabolism in early ocean alkaline hydrothermal systems, *Sci. Adv.*, 2019; 5: eaav7848.

J. Lee, A. Yamaguchi, H. Ooka, T. Kazami, M. Miyauchi, N. Kitadai, R. Nakamura, In situ FTIR study of CO₂ reduction on inorganic analogues of carbon monoxide dehydrogenase, *Chem. Commun.*, 2021, 57, 3267-3270

Probably, it is demanding, but it is worth investigating the catalytic activity of Ni compounds, such as H₂ oxidation. Or, the authors better add a few sentences about the perspective of periotic catalysis potentially driven by Ni-organo compounds.

Reviewer #2 (Remarks to the Author):

Kopplin 2023 review Communications Chemistry manuscript COMMSCHEM-23-0525-T

This paper is a continuation of a previous one published this year, cited in the current paper as reference

11. Both papers use the same prebiotic chemistry arena and focus on acetylene as a precursor for the synthesis of more complex organic molecules, with a key mention of acetaldehyde, a compound formed by hydration of acetylene. A key difference between the older paper and the new one is that in the former the catalyst is the inorganic molecule nickel sulfide, while the latter points to an organic complex, nickel atom chelated by one or more dithiolene moieties, as having a role in the emergence of acetaldehyde and other organic compounds in their reactor. The authors cautiously say the organic complexes “mechanistically characterize the role of nickel sulfide in catalyzing the formation of acetaldehyde” for lack of direct evidence for catalytic formation, I would dare suggest the dithiolene complexes may well be the catalysts for adding water molecules to acetylene in forming acetaldehyde. This may be justified by the well-known fact, mentioned several times in the present paper, that bacterial acetylene hydratases have as cofactor a pyranopterin metal complex, for which the direct metal chelating entity is a dithiolene moiety. The idea that acetylene is a precursor for a catalyst synthesis (well evidenced in the paper) and the substrate for the catalyzed reaction, further enhances the originality and the strength of the present paper. This adds to a central conclusion of the paper, that its experimental result point to a key role for acetylene as a crucial precursor for prebiotic organic syntheses and of dithiolene as an early organic catalyst in this arena.

Comments:

- 1) There is sufficient overlap between the two recent papers to warrant a continuity-addressing text in the introduction and discussion, providing in more detail what was presented in the previous paper (Ref 11 in the present paper), how it serves as basis for the new paper, and what are the current points of innovation.
- 2) As per what is suggested in the first paragraph of this review, I recommend addressing the progress made in understanding how nickel compounds affect the production of acetaldehyde and perhaps other organic molecules in a system that contains NiS and acetylene. Catalysis by complexes or “formation of acetaldehyde through the subsequent degradation of these organo-metal complexes” which is left a bit obscure.
- 3) I strongly suggest to have an added systematic figure that will clearly show the structures of all the compounds addressed in the paper, e.g. dithiolene, a typical dithiolene Nickel complex, pyranopterin etc and if preferred, give each an identification number for the text. Also, the reactions invoked in the paper, e.g. $4\text{NiS} + 2\text{C}_2\text{H}_2 \rightarrow (\text{C}_2\text{H}_2\text{S}_2)_2\text{Ni} + 3\text{Ni}$ (what happens with the free Ni?). In this vein, it will help to justify why the nickel complexes in Fig.2 are shown as if the ring is aromatic (circle inside the pentamer) and Fig. 3 shows the ring with c-c double bond.
- 4) It will be to have a sentence in the introduction about previous knowledge on the level of significance of acetylene in the realm of prebiotic chemistry.
- 5) The cofactors of bacterial acetylene hydratase have molybdenum and tungsten as the chelated metal. He authors justify studying nickel instead is that Ni is X10 more prevalent on earth than the other two elements. It will be good to provide reference for this fact, use reference 1 below to mention that nickel is in active sites of 9 contemporary enzymes (known so far), and discuss expectations on what would happen if Mo and W was used instead of Ni in this series of experiments.
- 6) The clause beginning with “The near insolubility of the isolated compound at room temperature...” seems weakly relevant to the main line of the paper, and should best go to the supporting file.
- 7) In Figure 3, is there a way to estimate the structures based on a database of such compounds? In the same vein, is there a guess for the structure of complexes with odd count of sulfur atoms?

8) The clause including the text "In the degradation process, nickel sulfide is consistently formed. The identification was made by observing an insoluble black solid in the reaction vessel. However, likely, the nickel and sulfur released during degradation precipitated at least partially into nickel sulfide. I find this obscure. First because reporting that when the central reaction product of this paper degrades, it goes back to one of the reactants (NiS) is not highly interesting. Second, the very last sentence requires decipherment.

9) In the Introduction, the authors state that "Our experimental setup finds its roots in the "Iron-sulfur world" theory of Wächtershäuser, who considered nickel or iron sulfide salts as catalytically active species... which allows surface catalysis. The authors then rightly claim that they "provide the first mechanistic evidence of abiogenic acetylene processing based on nickel bis(dithiolene)... allowing early acetaldehyde formation from acetylene in a prebiotic environment. Both "mineral surface catalysis" and "prebiotic environment" are terms for very early stages in life's origin.

The significance of this paper for understanding the origin of life could be appreciably enhanced if taking advantage of a crucial clause in the Discussion: "The described transportation role of the complex is also worth mentioning. The change in polarity, from an ionic form into a highly lipophilic organic form, could also lead to enhanced complex uptake into vesicles with less permeable and hydrophobic properties".

This suggests that the chemical characteristics of the studied catalytic metal complex facilitates its possible entry of an effective pro-enzyme into a vesicular protocell, a much more advanced stage en route to living cells. I strongly recommend adding text that states how the reported results are relevant not only to early syntheses on rocks or in the aqueous oceanic bulk, but also in evolving metabolism within protocells, the bridge between abiogenesis and life (may cite Refs 2,3 below).

Minor comments:

1) In a paper about life's origin, a discipline that uses "evolve" in another context, in sentences like "Its temporal evolution and possible function in this simulated early Earth atmosphere" it is strongly advisable to change to "temporal progression".

2) The clause including "The maximum at 500 nm explained the observed pink" leaves obscure what is the species in the system discussed that absorbs light at 500 nm and later degrades.

References

1. Alfano M and Cavazza C, Structure, function, and biosynthesis of nickel-dependent enzymes, *Protein Science* 29:1071–1089, (2020).
2. Nunes Palmeira R, Colnaghi, M, Harrison SA, Pomiankowski A, Lane N. The limits of metabolic heredity in protocells. *Proc. R. Soc. B* (2022)
3. Kahana, A and Lancet, D. Self-reproducing catalytic micelles as early nanoscopic protocells, *Nature Rev Chem* (2021)

Reviewer #3 (Remarks to the Author):

I enjoyed this paper very much, and found the study well conceived and validated. I recommend it for publication after fixing a few small issues:

1. I think the word ferment in the phrase "ferment acetylene to acetaldehyde" is incorrect, a better word would be "transform" or simply "hydrate". This occurs once in the second paragraph of the introduction, and once in the abstract.
2. On line 256, I think the word "motif" is intended by "motive."

3. Perhaps the sentence ending on Line 259 could supply some references?
4. On line 262, I think the reference is Begum et al.?
5. There is some inconsistency in the references, for example ref 4 includes the first author's first name, ref 9 lacks author names completely

Reviewers' comments:

Reviewer #1 (Remarks to the Author):

The paper by Diederich et al. reported the in-situ generation of Ni organo-compounds under simulated early atmospheric-ocean conditions. The generated products have a similar structure to enzymes, suggesting the geological origin of the Ni-containing enzyme. All the experiments, especially the characterization of the mixed products, were carefully performed and the interpretation of the data were solid. Considering the versatile catalytic and redox functions of Ni organo-compounds, the proposed model of abiotic formation of Ni catalysts is intriguing and will provide molecular basis understanding of the transition from geochemistry to more sophisticated chemical reactions catalyzed by proto-enzyme-like catalysts. This work also provides the new scientific bridge between pure inorganic chemistry in the laboratory and the prebiotic chemistry that drives the emergence of life on Earth. Given the novelty of this study, I support its publication in Communications Chemistry.

The authors thank the reviewer for the positive comments and the detailed analysis of our results.

A weakness of this paper is the geological condition on which the authors stand. The authors refer to the iron-sulfur world hypothesis, which is the idea that metal-sulfide minerals at deep-sea hydrothermal vents played a crucial role in prebiotic chemistry. Meanwhile, the author's research also stands on atmospheric photochemistry to generate CO and acetylene. Considering the diffusion processes of small molecules, I found it difficult to consider that CO and acetylene are transported from the atmosphere to the deep-sea hydrothermal vents against the dilution processes in air and water. In this regard, the authors may consider the recent works on Ni-sulfide catalyzed CO₂ to CO conversion and subsequent formation of prebiotically important compounds at deep-sea hydrothermal vents.

We agree with the reviewer's concerns and comments. As the reviewer states, our initial research idea indeed stems from the iron-sulfur world hypothesis. Nevertheless, we do not only consider deep-sea hydrothermal vents as the sole possible environment. We specifically want to refrain from defining a precise environment for the shown reaction, as prebiotic conditions, in general, are an ongoing topic of research. We added Ref 10 to the introduction, referencing a postulated hydrothermal surface-water environment on earth with acetylene and sulfide minerals. Our research focuses on the transition from geochemistry to proto-enzymes, as mentioned by the reviewer, but we aim to keep the question of a particular prebiotic environment open without making any definitive conclusions. However, the references suggested by the reviewer describe very interesting roles of nickel sulfide and present relevant chemistries that need to be mentioned in the present paper. Accordingly, we included the references and expanded the Introduction section as follows:

Line 96: "Nickel sulfide plays a crucial role in recent prebiotic chemistry, as they were shown to promote multiple relevant reactions. Thioesters were synthesized on nickel sulfide through geoelectrochemical CO₂ fixation²². Another study reported the promotion of a protometabolism in an alkaline hydrothermal vent environment²³."

N. Kitadai, R. Nakamura, M. Yamamoto, S. Okada, W. Takahagi, Y. Nakano, Y. Takahashi, K. Takai, Y. Oono, Thioester synthesis through geoelectrochemical CO₂ fixation on Ni sulfides, Commun. Chem., 2021, 4, 37.

N. Kitadai, R. Nakamura, M. Yamamoto, K. Takai, N. Yoshida, Y. Oono, Metals likely promoted protometabolism in early ocean alkaline hydrothermal systems, Sci. Adv., 2019; 5: eaav7848.

J. Lee, A. Yamaguchi, H. Ooka, T. Kazami, M. Miyauchi, N. Kitadai, R. Nakamura, In situ FTIR study of CO₂ reduction on inorganic analogues of carbon monoxide dehydrogenase, Chem. Commun., 2021, 57, 3267-3270.

Probably, it is demanding, but it is worth investigating the catalytic activity of Ni compounds, such as H₂ oxidation. Or, the authors better add a few sentences about the perspective of periotic catalysis potentially driven by Ni-organo compounds.

We completely agree with the reviewer that H₂-oxidation is a very intriguing reaction potentially catalyzed by the reported nickel-organic compound. This reaction will be investigated in a future study.

We added the following sentence to the introduction to highlight reactions of interest potentially catalyzed by Ni-organo compounds:

Line 100: "Ni-organo catalysts are pivotal in various organic reactions, including cross-coupling reactions (e.g., Suzuki, Kumada, or Negishi couplings)²⁴, carbon-heteroatom bond-forming²⁵ reactions and the hydrogenation of unsaturated organic compounds²⁶."

Reviewer #2 (Remarks to the Author):

Kopplin 2023 review Communications Chemistry manuscript COMMSCHEM-23-0525-T
This paper is a continuation of a previous one published this year, cited in the current paper as reference 11. Both papers use the same prebiotic chemistry arena and focus on acetylene as a precursor for the synthesis of more complex organic molecules, with a key mention of acetaldehyde, a compound formed by hydration of acetylene. A key difference between the older paper and the new one is that in the former the catalyst is the inorganic molecule nickel sulfide, while the latter points to an organic complex, nickel atom chelated by one or more dithiolene moieties, as having a role in the emergence of acetaldehyde and other organic compounds in their reactor. The authors cautiously say the organic complexes “mechanistically characterize the role of nickel sulfide in catalyzing the formation of acetaldehyde” for lack of direct evidence for catalytic formation, I would dare suggest the dithiolene complexes may well be the catalysts for adding water molecules to acetylene in forming acetaldehyde. This may be justified by the well-known fact, mentioned several times in the present paper, that bacterial acetylene hydratases have as cofactor a pyranopterin metal complex, for which the direct metal chelating entity is a dithiolene moiety. The idea that acetylene is a precursor for a catalyst synthesis (well evidenced in the paper) and the substrate for the catalyzed reaction, further enhances the originality and the strength of the present paper. This adds to a central conclusion of the paper, that its experimental result point to a key role for acetylene as a crucial precursor for prebiotic organic syntheses and of dithiolene as an early organic catalyst in this arena.

The authors thank the reviewer for the detailed assessment of our research and the interesting comments, ultimately improving our results.

Comments:

1) There is sufficient overlap between the two recent papers to warrant a continuity-addressing text in the introduction and discussion, providing in more detail what was presented in the previous paper (Ref 11 in the present paper), how it serves as basis for the new paper, and what are the current points of innovation.

We added the following continuity-addressing text in the introduction and thank the reviewer for this suggestion:

Line 106: “We now further investigated the previously mentioned one-pot setup¹⁴ with a focus on the formation mechanism of acetaldehyde and its link to the presence of nickel-organo compounds. The one-pot setup already showed a progressing system that generated acetaldehyde over time and also changed its pH in the process due to formed acids, ultimately leading to a changed reaction path for the formed acetaldehyde. Building on the knowledge about the presence of acetaldehyde, we provided the first mechanistic evidence of abiogenic acetylene processing based on nickel bis(dithiolene), a molecular structure reminiscent of active centers of contemporary enzymes. Considering the resemblance with molybdopterin coenzymes, we postulate that nickel bis(dithiolene) acted as a transition catalyzer, allowing early acetaldehyde formation from acetylene in a prebiotic environment.”

2) As per what is suggested in the first paragraph of this review, I recommend addressing the progress made in understanding how nickel compounds affect the production of acetaldehyde and perhaps other organic molecules in a system that contains NiS and acetylene. Catalysis by complexes or “formation of acetaldehyde through the subsequent degradation of these organo-metal complexes” which is left a bit obscure.

We added an additional paragraph to the discussion addressing this comment:

Line 327: “Our findings revealed a pathway from acetylene to acetaldehyde via a nickel-organo complex. This discovery strengthens the relevance of nickel-organo compounds for the origin-of-life, as they now present an intermediate product for a prebiotic synthesis pathway of acetaldehyde, higher aldehydes, the amino acid alanine, and potentially higher amino acids. Furthermore, this finding now paves the way for investigations about the exact catalytic nature and other roles in prebiotic chemistry of this nickel-organo compound.”

3) I strongly suggest to have an added systematic figure that will clearly show the structures of all the compounds addressed in the paper, e.g. dithiolene, a typical dithiolene Nickel complex, pyranopterin etc and if preferred, give each an identification number for the text. Also, the reactions invoked in the paper, e.g. $4NiS+2C_2H_2 \rightarrow (C_2H_2S_2)_2Ni+3Ni$ (what happens with the free Ni?). In this vein, it will help to justify why the nickel complexes in Fig.2 are shown as if the ring is aromatic (circle inside the pentamer) and Fig. 3 shows the ring with c-c double bond.

We agree and add a systematic figure to S.I. that shows the structures of all reactants, a typical dithiolene nickel complex, reaction products, and the pterin co-factor. Dithiolene itself is not shown as it is not detected in the system, only as the nickel complex. We, therefore, ignore if a free dithiolene species exists in the system. We want to refrain from adding the identification numbers as the number of completely identified compounds in the paper is very low. We mentioned the supplementary figure in the main text as follows:

Line 173: “A reaction scheme and discussed compounds can be found in the supplementary information (Suppl. Figure 2).”

Supplementary Figure 1: Reaction scheme overview. Panel A shows the hypothesized formation pathway of nickel bis(dithiolene) and acetaldehyde. Panel B shows the structure of a molybdopterin co-factor found in extant enzymes. *Unknown form.

The question about the remaining nickel is interesting and we do not have a precise answer for this question. Free Ni²⁺ ions can be precipitated from the reaction mixture at later points in the time series, as discussed in reference 14. This would also be the most straightforward hypothesis for this reaction.

We agree with the reviewer and change the aromatic ring into a double bond in Figure 2 to stay consistent throughout the manuscript.

4) It will be to have a sentence in the introduction about previous knowledge on the level of significance of acetylene in the realm of prebiotic chemistry.

We agree and add the following sentence:

Line 72: "Previous hypotheses regarding the role of acetylene in the origin of life have suggested that it contributes to the formation of polycyclic aromatic hydrocarbons¹¹ and small biomolecules¹². Its popularity increased after it was proposed as a "fast food" for organisms in Earth's early biosphere⁹."

5) The cofactors of bacterial acetylene hydratase have molybdenum and tungsten as the chelated metal. The authors justify studying nickel instead is that Ni is X10 more prevalent on earth than the other two elements. It will be good to provide reference for this fact, use reference 1 below to mention that nickel is in active sites of 9 contemporary enzymes (known so far), and discuss expectations on what would happen if Mo and W was used instead of Ni in this series of experiments.

We thank the reviewer for this comment and added the following information provided in reference 37:

Line 321: "Yet nickel is a much more frequent element on Earth with an elemental abundance of 1.86% compared to Mo with 2.35 ppm and W with 180 ppb."

We added the suggested reference with the sentence:

Line 323: "Additionally, nickel is found in the active centers of 9 contemporary enzymes (known so far). Further studies have to be performed to investigate what prebiotic conditions allow for an exchange of nickel through molybdenum or tungsten."

6) The clause beginning with "The near insolubility of the isolated compound at room temperature..." seems weakly relevant to the main line of the paper, and should best go to the supporting file.

Even though we see the reviewer's point, we prefer to keep this information in the main text, as it is essential for the discussion about the change in the hydrophobicity of the compound. As pointed out later by the reviewer, the potential idea to merge through lipophilic membranes is an interesting part of the discussion and should, therefore, be strengthened by data in the main text.

7) In Figure 3, is there a way to estimate the structures based on a database of such compounds? In the same vein, is there a guess for the structure of complexes with odd count of sulfur atoms?

We are not aware of any database containing such structures. We were unable to find plausible structures for the annotations with odd amounts of sulfur. Those species need further investigation in future studies to understand their formation and structure. Some of them are likely to be in-source fragments, as already stated in the text.

8) The clause including the text “In the degradation process, nickel sulfide is consistently formed. The identification was made by observing an insoluble black solid in the reaction vessel. However, likely, the nickel and sulfur released during degradation precipitated at least partially into nickel sulfide. I find this obscure. First because reporting that when the central reaction product of this paper degrades, it goes back to one of the reactants (NiS) is not highly interesting. Second, the very last sentence requires decipherment.

This observation is the basis for the part of the discussion starting at line 267 and should, therefore, be strengthened by data in the main text:

Line 267: “The described transportation role of the complex is also worth mentioning. The ability to extract nickel sulfide from minerals, to be later redeposited after degradation, leads to new scenarios, where nickel-catalyzed reactions like the reduction of double bonds²⁹ can occur on primitive Earth.”

The last sentence has been modified to enhance its clarity and ease of understanding:

Line 227: “This particular process provides the complex with the ability to dissolve nickel sulfide, making it soluble, transport this highly insoluble salt, and relocate it via diffusion processes.”

9) In the Introduction, the authors state that “Our experimental setup finds its roots in the “Iron-sulfur world” theory of Wächtershäuser, who considered nickel or iron sulfide salts as catalytically active species... which allows surface catalysis. The authors then rightly claim that they “provide the first mechanistic evidence of abiogenic acetylene processing based on nickel bis(dithiolene)... allowing early acetaldehyde formation from acetylene in a prebiotic environment. Both “mineral surface catalysis” and “prebiotic environment” are terms for very early stages in life’s origin. The significance of this paper for understanding the origin of life could be appreciably enhanced if taking advantage of a crucial clause in the Discussion: “The described transportation role of the complex is also worth mentioning. The change in polarity, from an ionic form into a highly lipophilic organic form, could also lead to enhanced complex uptake into vesicles with less permeable and hydrophobic properties”. This suggests that the chemical characteristics of the studied catalytic metal complex facilitates its possible entry of an effective pro-enzyme into a vesicular protocell, a much more advanced stage en route to living cells. I strongly recommend adding text that states how the reported results are relevant not only to early syntheses on rocks or in the aqueous oceanic bulk, but also in evolving metabolism within protocells, the bridge between abiogenesis and life (may cite Refs 2,3 below).

We added the following text to the discussion to address this highly constructive comment:

Line 272: “This aspect makes the reported complex not only relevant for early prebiotic stages but also for later stages during a transition from reactions in an aqueous bulk to an evolving metabolism in protocells. Its chemical nature facilitates its introduction into vesicular protocells. This hypothesis is strengthened by a recently published report of vesicle formation in the described set-up, where the formed long-chain fatty acids showed the potential to form vesicles³⁰.”

Minor

comments:

1) In a paper about life’s origin, a discipline that uses “evolve” in another context, in sentences like “Its temporal evolution and possible function in this simulated early Earth atmosphere” it is strongly advisable to change to “temporal progression”.

Based on the reviewer's feedback, we have carefully considered and revised our wording to align with the suggested formulation. Thank you for bringing this to our attention.

2) The clause including "The maximum at 500 nm explained the observed pink" leaves obscure what is the species in the system discussed that absorbs light at 500 nm and later degrades.

We agree that further clarification is needed and have revised the sentence accordingly:

Line 148: "UV/VIS spectroscopy of the colored reaction product in water showed two absorption maxima at 500 and 855 nm. The second maximum is in the near-infrared at 855 nm. Absorption in the near-infrared is characteristic of bis(dithiolene)nickel compounds²⁷. The maximum at 500 nm explained the observed pink color as absorption of green light results in red transmitted light and is also expected to be associated with the detected complex."

Reviewer #3 (Remarks to the Author):

I enjoyed this paper very much, and found the study well conceived and validated. I recommend it for publication after fixing a few small issues:

We thank the reviewer for the very positive feedback and evaluation of our work.

1. I think the word ferment in the phrase "ferment acetylene to acetaldehyde" is incorrect, a better word would be "transform" or simply "hydrate". This occurs once in the second paragraph of the introduction, and once in the abstract.

We agreed and changed the wording as recommended from fermented to "hydrated".

2. On line 256, I think the word "motif" is intended by "motive."

We have corrected the error and would like to express our gratitude to the reviewer for their careful reading.

3. Perhaps the sentence ending on Line 259 could supply some references?

We prefer to leave the sentence without references to prevent redundancy. Indeed, the following sentences discuss the catalyzed reactions in more detail, including the corresponding, specific references 6+13+31.

4. On line 262, I think the reference is Begum et al.?

We agree and added this reference to line 286, which was initially cited as Ref. 6

5. There is some inconsistency in the references, for example ref 4 includes the first author's first name, ref 9 lacks author names completely

We thank the reviewer for this observation, corrected the inconsistencies in ref. 4 and ref. 9 and updated the other references.

REVIEWERS' COMMENTS:

Reviewer #1 (Remarks to the Author):

The authors have responded appropriately to all my comments.
I recommend that this paper be accepted as is.

Reviewer #2 (Remarks to the Author):

I am content, approve to publish.